# Aggressive Prostate Cancer in Patients Treated with Active Surveillance

**DOI:** 10.3390/cancers15174270

**Published:** 2023-08-25

**Authors:** Yoichiro Tohi, Takuma Kato, Mikio Sugimoto

**Affiliations:** Department of Urology, Faculty of Medicine, Kagawa University, Kagawa 761-0793, Japan

**Keywords:** active surveillance, aggressive prostate cancer, low-risk prostate cancer, favorable intermediate-risk prostate cancer

## Abstract

**Simple Summary:**

Active surveillance is a strategy used to manage early-stage prostate cancer without immediate treatment, aiming to maintain patients’ quality of life. However, concerns remain about identifying “aggressive prostate cancer” within the active surveillance cohort, which refers to cancers with a higher potential for progression. We defined aggressive prostate cancer within the active surveillance cohort as follows: a cancer that is not an indication for active surveillance, despite low-risk or favorable intermediate-risk prostate cancer, and will undergo pathological upgrading during the course of active surveillance monitoring. The identification of aggressive prostate cancer during active surveillance is crucial, as it indicates the need for a shift in the treatment strategy. The detection of aggressive prostate cancer in these cohorts enables a timely intervention and the initiation of appropriate definitive treatment options to improve patient outcomes. To tackle this, previous studies have suggested a personalized follow-up approach that uses a mix of clinical data, biomarkers, and genetic factors to assess risk. As active surveillance indications expand, the importance of identifying aggressive prostate cancer through a personalized risk-based follow-up is expected to increase.

**Abstract:**

Active surveillance has emerged as a promising approach for managing low-risk and favorable intermediate-risk prostate cancer (PC), with the aim of minimizing overtreatment and maintaining the quality of life. However, concerns remain about identifying “aggressive prostate cancer” within the active surveillance cohort, which refers to cancers with a higher potential for progression. Previous studies are predictors of aggressive PC during active surveillance. To address this, a personalized risk-based follow-up approach that integrates clinical data, biomarkers, and genetic factors using risk calculators was proposed. This approach enables an efficient risk assessment and the early detection of disease progression, minimizes unnecessary interventions, and improves patient management and outcomes. As active surveillance indications expand, the importance of identifying aggressive PC through a personalized risk-based follow-up is expected to increase.

## 1. Introduction

Prostate cancer (PC) is one of the most prevalent malignancies among men and has a considerable impact on morbidity and mortality worldwide [1]. With an aging population, the number of patients with PC is expected to increase [2]. The widespread use of prostate-specific antigen (PSA) screening has increased PC diagnoses and has resulted in the early detection of less malignant and localized PC [3,4]. Less malignant and localized PC is an insignificant cancer and unnecessary interventions for this may lead to overtreatment. Definitive treatments such as a prostatectomy or radiotherapy for insignificant PC can decrease one’s quality of life (QOL) in terms of urinary, sexual, and bowel functions [5]. Therefore, the management of PC has evolved over the years, with active surveillance (AS) emerging as a promising approach for select patients with low-risk PC [6,7]. AS entails the close monitoring of cancer progression via regular PSA testing, digital rectal examinations (DREs), magnetic resonance imaging (MRI), and repeat prostate biopsies, while deferring immediate curative treatments [8,9,10]. This approach aims to minimize overtreatment and its associated adverse effects such as urinary incontinence and erectile dysfunction. In addition, AS potentially maintains the health-related QOL of the patient [11,12,13].

Although AS has gained acceptance and demonstrated favorable outcomes in several studies [14,15,16], concerns regarding the identification of “aggressive PC” within this cohort remain. “Aggressive PC” is defined as a subset of cancers that exhibit a higher potential for progression. These cancers can also affect patient outcomes.

Therefore, this review aimed to comprehensively examine the incidence, predictive factors, and appropriate management strategies for aggressive PC in patients undergoing AS. By evaluating the existing literature and discussing key studies in this area, we aim to provide a comprehensive overview of aggressive PC within AS and discuss the possibility of risk and the early detection of aggressive PC.

## 2. Overview of Active Surveillance

### 2.1. Principles of Active Surveillance

AS aims to identify patients whose cancers demonstrate indolent behavior and are unlikely to substantially progress during their lifetime, thereby allowing the patient to avoid unnecessary definitive treatments and their associated side effects. AS involves monitoring and timely curative treatment when disease progression is a concern. Thus, AS is a curative setting that differs from watchful waiting (WW) in this respect. WW is another strategy aimed at preventing overtreatment, in which palliative treatment is initiated to maintain the QOL only after disease progression, such as pain associated with bone metastases or the appearance of urgent clinical symptoms such as hematuria or a urinary tract obstruction. Thus, WW is a palliative treatment that does not consider the clinical stages of PC.

### 2.2. Inclusion Criteria of Active Surveillance

The selection of patients for AS requires the consideration of various factors, including age, PSA levels, the Gleason score, the clinical stage, the tumor volume, and patient preference. The inclusion criteria are as follows: low-grade, insignificant PC with a non-life-threatening prognosis. Table 1 shows the current recommendations of various guidelines, including those of the American Urological Association (AUA), the European Association of Urology (EAU), the National Comprehensive Cancer Network (NCCN), and the Japanese Urological Association [8,9,10,17]. 

The AUA guidelines state that AS is the only strong recommendation for low-risk PC. However, in the case of low-risk PC with a large tumor volume, a high PSA density, a family history of PC, or germline mutations, curative treatment may be an option after careful discussion with the patient [8]. For favorable intermediate-risk PC, the AUA guidelines strongly recommend that AS be offered as a treatment option along with a prostatectomy and radiation therapy, after explaining to the patient the lack of medium- to long-term evidence and the risk of metastases occurring during AS [8]. 

According to the EAU guidelines, AS is a strong recommendation for low-risk PC with an expected life expectancy of 10 years or more, and WW is strongly recommended for low-risk PC with an expected life expectancy of <10 years [9]. In addition, during AS induction, an MRI evaluation and a targeted biopsy at the time of the confirmatory biopsy are recommended for lesions of prostate imaging reporting and data system (PI-RADS) category 3 or higher [9,18]. The EAU guidelines state that AS can be considered for low-volume intermediate-risk PC (defined as <3 positive systematic cores and <50% core involvement) or other single elements of intermediate risk. Patients with intraductal carcinoma of the prostate (IDC-P), a cribriform pattern, sarcomatoid carcinoma, small-cell carcinoma, an extraprostatic extension, or a lymphovascular invasion on a prostate biopsy should be excluded [9]. 

The NCCN guidelines recommend AS for low-risk PC if the life expectancy is ≥10 years, and radiation therapy and a prostatectomy are also included if the life expectancy is 20 years. For low-risk PC, the NCCN recommends WW if the life expectancy is <10 years and AS if the life expectancy is ≥10 years, along with radiation therapy and a prostatectomy [10]. The NCCN guidelines also recommend AS for patients with favorable intermediate-risk PC [10]. 

Thus, AS started as an indication for low-risk PC; however, in recent years, it has been expanded to include some intermediate-risk PC cases that are considered to have a good prognosis.

### 2.3. Protocol of Active Surveillance

The AS protocol typically involves regular follow-up visits, which may include PSA testing, a DRE, MRI, and repeat prostate biopsies. The frequency of these tests may vary depending on the individual patient characteristics and institutional protocols. Table 2 lists the follow-up protocols for various guidelines [8,9,10,17]. The purpose of these assessments is to evaluate the disease stability and promptly detect signs of disease progression. The trigger for definitive treatment is pathological progression based on the inclusion criteria of reclassification. Reclassification is often defined by a Gleason score upgrading or an increase in cancer volume. Some patients on AS are anxious because of their cancer-carrying status [19,20]; therefore, a definitive treatment may be reasonable for patients under AS owing to the patient’s anxiety. 

## 3. Aggressive Prostate Cancer during Active Surveillance

### 3.1. Definition of Aggressive Prostate Cancer during Active Surveillance 

Aggressive PC can be defined as a cancer that is not an indication for AS, despite low-risk or favorable intermediate-risk PC, and will undergo pathological upgrading during the course of AS monitoring. Pathological upgrading refers to a change in the characteristics of the disease status upon a reevaluation through repeat biopsies, revealing a higher Gleason score or a more extensive disease than initially diagnosed. Pathological upgrading is referred to as reclassification in AS. 

The identification of aggressive PC during AS is crucial, as it indicates the need for a shift in the treatment strategy. Upgraded PC may exhibit a greater potential for adverse clinical outcomes. The detection of aggressive PC in these cohorts enables a timely intervention and the initiation of appropriate definitive treatment options to improve patient outcomes.

### 3.2. Definition of Pathological Reclassification

Currently, various protocols exist; Table 3 lists the definitions of reclassification [14,16,21,22,23,24]. Each protocol is based on Gleason score upgrading, upstaging, an increased tumor volume, and PSA kinetics. The prostate cancer research international active surveillance (PRIAS) protocol focuses on IDC-P and cribriform patterns in repeat biopsies [24]. The presence of these features in biopsy specimens triggers definitive treatment [24].

The use of PSA kinetics as the sole trigger for definitive interventions in AS protocols has been debated [25,26,27,28,29]. Limited evidence suggests that elevated pretreatment PSA levels are associated with an adverse pathology during a radical prostatectomy (RP) or subsequent biochemical recurrence after definitive treatment [30,31]. Several studies, including one conducted by the Johns Hopkins group, have shown that the PSA doubling time (PSA-DT) and velocity are not reliable predictors of pathological progression in repeat biopsies [25]. There is no specific threshold for the PSA velocity or doubling time, which have both a high sensitivity and specificity for detecting progression in repeat biopsies [25].

**Table 3 cancers-15-04270-t003:** Reclassification and the rate of definitive treatment during AS.

Research Group	Inclusion Criteria					Trigger for Definitive Treatment	Rate of Reclassification	Definitive Treatment Rate
	Clinical Stage	Gleason Score	Cancer Volume	PSA (ng/mL)	PSA Density (ng/mL/cm^3^)			
John Hopkins University	T1c	6	≤2 cores positive, ≤50% core involvement per core	-	≤0.15	GS ≥3 + 4, 3 positive cores, >50% cancer per core	26% at 10 years, 31% at 15 years [16]	50% at 10 years, 57% at 15 years [16]
University of Toronto	≤T2a	3 + 3 or 3 + 4 if aged >70 years	-	≤10 or ≤15 if aged >70 years	-	GS upgrading, clinical progression	25.6% (median follow-up: 6.4 years) [14]	27% (median follow-up: 6.4 years) [14]
Memorial Sloan Kettering Cancer Center	≤T2a	6	≤2 cores positive, ≤50% core involvement per core	≤10	-	GS upgrading, clinical progression, upstaging	24% at 5 years, 36% at 10 years, 41% at 15 years [21]	24% at 5 years, 36% at 10 years, 42% at 15 years [21]
University of California, San Francisco	≤T2a	6	<33% positive cores	≤10	-	GS ≥ 3 + 4, PSA velocity >0.75	60% at 7 years [22]	41% at 7 years [22]
Canary Prostate Active Surveillance Study	≤T2c	6 and 3 + 4	-	-	-	GS upgrading, clinical progression, PSA-DT < 3 year	22.4% (median follow-up: 5.6 years) [32,33]	32.5% (median follow-up: 5.6 years) [32,33]
PRIAS	≤T2	≤6	no limitation if MRI available at diagnosis (if not, ≤2 cores positive)	≤20 (if MRI available at diagnosis), ≤10 (if MRI not available)	<0.25 if MRI available at diagnosis (if not, ≤0.2)	progression to ≥T3, GS ≥ 4+3, or GS≤ 3 + 4 with IDC-P/cribriform	34% at 5 years, 41% at 10 years [15]	20.8% (median follow-up: not available) [15]
		3 + 4	≤50% positive cores if MRI available at diagnosis (if not, ≤2 cores positive)	≤20 (if MRI available at diagnosis), ≤10 (if MRI not available)	<0.25 if MRI available at diagnosis (if not, ≤0.2)			

PSA, prostate-specific antigen; PRIAS, prostate cancer research international active surveillance; IDC-P, intraductal carcinoma of the prostate; GS, Gleason score.

### 3.3. Rate of Reclassification and Definitive Treatment during Active Surveillance

Evidence from multiple studies has highlighted the occurrence of reclassification and the rate of definitive treatment during AS (Table 3).

According to a report from Johns Hopkins University, the study had a median follow-up period of five years, and the cumulative incidence of grade reclassification, which was a Gleason score ≥ 3 + 4, three positive biopsy cores, and >50% cancer per core, was 26% at 10 years and increased to 31% at 15 years. Similarly, the cumulative incidence of a curative intervention was 50% at 10 years and increased to 57% at 15 years [16]. According to a report from the University of Toronto, this study had a median follow-up period of 6.4 years. The definitive treatment rate, which was determined by reclassification factors such as a short PSA-DT, grade progression, stage progression, and biopsy volume progression, was 25.6%. The most common reasons for initiating treatment were the PSA-DT and grade progression. The definitive treatment rate was 27% [14]. Findings from the Memorial Sloan Kettering Cancer Center revealed that, by defining triggers such as Gleason score upgrading, clinical progression, and upstaging, the study reported reclassification rates of 24, 36, and 41% at 5, 10, and 15 years, respectively [21]. The University of California, San Francisco, defined the triggers for definitive treatment as a Gleason score ≥3 + 4 and a PSA increase >0.75 ng/mL per year. The study reported a reclassification rate of 60% at seven years and a definitive treatment rate of 41% at the same time point [22]. According to the Canary Prostate Active Surveillance Study report, the triggers for definitive treatment were Gleason score upgrading, clinical progression, and a PSA-DT < 3 years. The study had a median follow-up of 5.6 years, with a reclassification rate of 22.4% and a definitive treatment rate of 32.5% [32,33]. In the report from PRIAS, the triggers for definitive treatment were defined as progression to ≥T3, a Gleason score ≥ 4 + 3, or a Gleason score ≤ 3 + 4 with IDC-P or cribriform features. The rate of reclassification was 34% at five years and 41% at 10 years, with a definitive treatment rate during the study period of 20.8% [15].

## 4. Risk Factors for Aggressive Prostate Cancer during Active Surveillance

### 4.1. Aggressive Prostate Cancer That Is Not an Indication for Active Surveillance, despite Low-Risk or Favorable Intermediate-Risk PC

AS is for low-grade and localized PC that is not life-threatening. However, certain pathological factors may preclude AS and require immediate treatment. These factors are indicative of aggressive PC and raise concerns regarding disease progression and adverse clinical outcomes.

#### 4.1.1. Tumor Volume

Several AS protocols have emerged whose inclusion criteria are in accordance with the Epstein criteria [34]: a clinical stage of T1c, a PSA density <0.15 ng/mL, no Gleason pattern 4 or 5, <3 positive cores, and <50% cancer per core. These criteria indicate a low-grade, low-volume PC. Therefore, certain cases with considerable tumor volumes may warrant immediate treatment owing to the increased risk of disease progression and adverse outcomes. Several studies have demonstrated an association between larger tumor volumes and aggressive disease characteristics [35,36]. A multi-institutional cohort of 7279 patients with AS showed that one of the factors associated with an earlier conversion to definitive treatment was a greater number of cancerous biopsy cores, and patients with high-volume tumors in the lowest Gleason grade group had a shorter time to conversion and exhibited similar behavior to higher-risk patients [35]. A study on a cohort of 651 men (144 with intermediate-risk PC) for a median of 4.5 years showed that the percentage of positive biopsy cores and the presence of Gleason pattern 4 on the initial biopsies were independent predictors of disease progression in AS [36]. The decision to exclude cases with a high tumor volume from AS is based on the understanding that larger tumors have a greater potential for disease progression. 

#### 4.1.2. IDC-P and Cribriform Pattern

The presence of IDC-P or cribriform patterns results in poor oncological outcomes concerning biochemical recurrence, metastasis, and disease-specific mortality after a prostatectomy [37,38,39,40]. In studies using prostate specimens, the presence of IDC-P is significantly associated with an advanced disease stage and higher postoperative PSA recurrence rates [41]. Moreover, patients with IDC-P are resistant to hormonal and radiation therapies [42,43]. The prevalence of IDC-P in RP specimens with a Gleason score of 3 + 3 or 3 + 4, which indicates patients that may be candidates for AS, is 4%, particularly when Gleason pattern 4 is below 10% in cases with a Gleason score of 3 + 4 [40]. RP specimens of upfront AS showed that IDC-P or a cribriform RP comprised approximately one-third of all RP specimens in men who underwent an RP following AS, indicating lower PSA recurrence-free survival [44]. In particular, IDC-P—which has gained popularity as a prognostic factor—is independent of the Gleason classification and should be detected with caution. 

Gleason pattern 4 is highly heterogeneous; among these, the tumor prognosis of the cribriform pattern is particularly unfavorable. According to a report analyzing the tumor prognosis of each type of Gleason pattern 4 present in RP specimens with a Gleason score of 7, the presence of a cribriform pattern within Gleason pattern 4 is a prognostic factor for both metastasis-free and cancer-specific survival [45]. Specifically, the presence of a large cribriform pattern (more than twice the size of the surrounding normal glands) diagnosed as a Gleason score of 3 + 4 in RP specimens is also associated with a poor prognosis [46].

Initially, AS was reserved for low-risk PC with a Gleason score of 3 + 3; however, the eligibility criteria have relaxed in recent years for some intermediate-risk PC cases. The motivation behind expanding the eligibility criteria is based on the finding that PC patients with a Gleason score of 3 + 4, excluding those with IDC-P or a cribriform pattern, exhibit oncological outcomes following definitive treatment comparable to those of patients with a Gleason score of 3 + 3 [38,41]. 

#### 4.1.3. Percentage of Gleason Pattern 4 in Prostate Biopsy Cores

A low volume of Gleason pattern 4 in prostate biopsies is associated with favorable pathological findings in RP specimens and good postoperative oncological outcomes [47,48,49,50]. A study analyzing the correlation between the percentage of Gleason pattern 4 in prostate biopsy cores and the pathological findings in RP specimens revealed that patients with a Gleason score of 3 + 4 and Gleason pattern 4 occupying <5% of the biopsy cores had a Gleason score at the RP, a pT stage, a total tumor volume, and an insignificant cancer rate similar to patients with a Gleason score of 3 + 3 [47]. Alternatively, in patients with a Gleason score of 3 + 4 and Gleason pattern 4 occupying 6–50% of the biopsy cores, the pathological findings in prostatectomy specimens were significantly worse than those with a Gleason score of 3 + 4 in <5% of the biopsy cores [47]. Furthermore, a study analyzing the correlation between the percentage of Gleason pattern 4 in prostate biopsy cores and the post-prostatectomy tumor prognosis reported a significant association between the percentage of Gleason pattern 4 in biopsy cores and postoperative PSA recurrence [48]. In a study involving 608 patients with low-volume intermediate-risk PC (one or two cores with a Gleason score of 3 + 4 and PSA levels <20 ng/mL) who underwent an RP, approximately 25% of the participants exhibited a Gleason score of ≥4 + 3, seminal vesicle invasion, or lymph node involvement [49]. One study suggested that patients with a Gleason score of 3 + 4 and <0–20% Gleason pattern 4 involvement may be suitable candidates for AS, whereas those with a Gleason score of 3 + 4 and >20% Gleason pattern 4 involvement or with a disease with a Gleason score of 4 + 3 should undergo treatment [50]. 

### 4.2. Aggressive Prostate Cancer Undergoes Pathological Upgrading during Active Surveillance Monitoring

In this context, aggressive PC indicates PC, which is a risk factor for pathological upgrading (reclassification) on a repeat biopsy during AS, or an adverse pathology in RP specimens after upfront AS.

#### 4.2.1. Risk Factors for Reclassification on Repeat Biopsies during Active Surveillance

Several studies have investigated the predictors of reclassification in patients on AS. Some studies have focused on identifying biomarkers [51,52,53,54,55,56]. Press et al., assessed the role of the Decipher genomic classifier in predicting biopsy upgrading in AS. An analysis of 133 patients revealed a significant association between the Decipher scores and biopsy upgrading, particularly in Gleason grade group 1 disease. Integrating the Decipher score improved the predictive ability of the clinical model [51]. Gandellini et al., analyzed circulating miRNAs to identify those associated with disease reclassification to improve risk refinement. A three-miRNA signature (miR-511-5p, miR-598-3p, and miR-199a-5p) predicted reclassification [52]. Lonergan et al., aimed to identify the predictors of biopsy reclassification in men on AS for low-risk PC. An analysis of 1031 patients revealed that a high genomic score, the PSA kinetics, and a PSA density ≥0.15 were associated with reclassification within 3 years. The PSA kinetics and density remained associated with reclassification at 5 years [53]. Newcomb et al., investigated the association of the urinary biomarkers PCA3 and TMPRSS2:ERG with biopsy reclassification in men on AS. Among the 782 participants, PCA3 was associated with short-term reclassification at the first surveillance biopsy; however, TMPRSS2:ERG showed no association [54]. Regarding the genetic markers, Carter et al., investigated the association between germline mutations in DNA repair genes (*BRCA1/2* and *ATM*) and grade reclassification in men undergoing AS. An analysis of the two independent cohorts revealed that mutation carriers had a higher risk of reclassification than non-carriers. Specifically, *BRCA2* mutations are associated with a higher risk of reclassification [55]. DNA repair genes, especially *BRCA2*, are considered a risk for reclassification during AS, and the recognition of this carrier is important in implementing AS. Kato et al., identified the predictive markers for AS reclassification. This study focused on p2PSA-related parameters, including the %p2PSA, the prostate health index (PHI), and their thresholds. The results showed that the %p2PSA and PHI before the 1-year protocol biopsy had a good predictive power [56]. Some studies have focused on imaging modalities [57,58,59,60]. MRI has been implemented as a standard of care in AS. Its diagnostic potential is expected to identify an increased volume of the index lesion or the onset of de novo lesions during AS. At this stage, however, it is not yet practical to omit repeat prostate biopsies and replace them entirely with MRI alone. The reason for this is that MRI is reported to detect a non-negligible percentage of significant cancers in regions with no abnormal findings [61]. In other words, MRI in combination with other parameters may contribute to omitting repeat prostate biopsies from AS. On the other hand, MRI has improved the visualization of lesions, such as the anterior prostate [62], and has reduced sampling errors and underdiagnoses during systematic biopsies. In fact, when prostate cancer is diagnosed by an MRI-fusion biopsy and AS is initiated, the rate of discontinuation of AS has been reported to be reduced [63]. Huang et al., evaluated the association between the apparent diffusion coefficient (ADC) values on initial multiparametric MRI (mpMRI) and biopsy grade reclassification to Gleason grade group ≥2 in men on AS with Gleason grade group 1 PC. These results demonstrated that lower ADC values in the index lesion are significantly associated with an increased risk of reclassification. Incorporating ADC measurements into multivariable risk-prediction tools may improve the risk stratification for patients undergoing AS [57]. Williams et al., assessed the association between bilateral PC at enrollment in AS and progression to grade group ≥2. The results revealed that a bilateral disease and a higher PSA density at the confirmatory biopsy were associated with an increased risk of progression [58]. Saout K. et al., reported that a PI-RADS ≥4 on MRI was associated with an increased risk of treatment during AS, between negative MRI and a PSA density ≤0.10 during follow-up, and had an excellent negative predictive value to predict treatment [59]. This indicates that the likelihood of reclassification is low if there are no findings on MRI and if the PSA density is low. Schwen et al., evaluated the combined predictive value of the PHI and mpMRI for reclassification in men on AS. Among the 253 patients on AS, higher PHI values, PHI densities, and PSA densities were observed in individuals with reclassification. The combination of the PHI and mpMRI can aid in predicting reclassification [60]. Dai et al., identified AS candidates who would benefit the most from confirmatory biopsies to minimize grade under-classification. A multivariable analysis revealed that increasing age and a higher number of positive cores at the time of the diagnosis were independent predictors of reclassification among 556 men diagnosed with a Gleason score of 3 + 3 [64]. Similarly, Druskin et al., reported an association between older age and reclassification [65]. De la Calle et al., evaluated the clinical implications of perineural invasion in men undergoing AS for Gleason grade group 1 PC. Among the 1969 patients, those with a biopsy-detected perineural invasion had higher rates of grade reclassification [66]. Some reports have focused on approaches of prostate biopsies to reduce the risk of these upgradings. Zattoni F. et al., reported that a transperineal targeted biopsy may improve the concordance of clinically significant PC, with the final pathology compared with a transrectal targeted biopsy in a large worldwide population [67]. Pepe P et al., reported that a saturation prostate biopsy had a higher detection rate for clinically significant PC at the time of the confirmatory biopsy for men under AS [68]. Recently, prostate-specific membrane antigen positron emission tomography (PSMA-PET) has attracted attention as a new imaging method for prostate cancer. Many reports have indicated that the diagnostic accuracy of PSMA-PET is superior to that of conventional imaging [69]. Pepe P. et al., reported that a PSMA-PET-assisted prostate biopsy demonstrated a good accuracy in the diagnosis of clinically significant PC that was not inferior to an mpMRI-assisted prostate biopsy [70].

#### 4.2.2. Risk Factors for Adverse Pathology on Radical Prostatectomy Specimens after Upfront Active Surveillance

The appearance of an adverse pathology in RP specimens is the primary prognostic endpoint. Several studies have investigated predictors of an adverse pathology in RP specimens after upfront AS. Here, an adverse pathology was defined as Gleason grade group 3 or greater, pathological T3 or greater, or pathological N positivity. Tosoian et al., conducted a retrospective analysis of a prospective AS registry, and reported that the rate of an adverse pathology (Gleason grade group 3 or greater, pT3b, or pN positivity) for RP specimens was 21.3%. The results showed that an older age, the PSA density, and the preoperative risk are associated with an adverse pathology [71]. Bokhorst et al., presented the 10-year follow-up data for men enrolled in the PRIAS study and showed that a Gleason score >6 on the last biopsy was associated with an adverse pathology (Gleason grade group 3 or greater, pathological T3 or greater) for RP specimens [15]. Björnebo et al., conducted a population-based cohort study of 6021 low-risk patients with PC and 37.7% had an adverse pathology (Gleason grade group 3 or greater, pathological T3 or greater) for their RP specimens. A clinical T-stage and the PSA density at the time of the diagnosis, the age, the PSA density at the last re-biopsy, and the PI-RADS at the last re-biopsy were significantly associated with AP [72]. De la Calle et al., assessed whether older men with Gleason grade group 1 PC on AS had a higher risk of adverse pathologies in a delayed RP. Among 365 patients, the incidence of an adverse pathology (Gleason grade group 3 or greater, pT3 or greater, or pN positivity) was 59.2 and 44.1% for those ≥65 years and <65 years, respectively. In the older age group, mpMRI was a predictor of an adverse pathology [73]. Marenghi et al., investigated adverse pathologies in 21,169 patients in the GAP3 cohort who underwent a delayed RP after AS. The rate of an adverse pathology (Gleason grade group 3 or greater, pT3 or greater, pN positivity, resection margin) was 39.2%. Age, the PSA density, and the number of positive cores were independently associated with APFs. AS remains a safe option for low-risk patients; however, the wider entry criteria are associated with certain adverse pathologies [74]. Tohi et al., examined the predictive factors associated with adverse pathologies in a prospective cohort enrolled in PRIAS-JAPAN. In this study, the incidence of adverse pathologies (Gleason grade group 3, pT3 or greater, pN positivity, or the presence of IDC-P or cribriform patterns) was 48.9%. Increasing age at the time of AS enrollment and before an RP were the only predictive factors for adverse pathologies [75]. 

## 5. Appropriate Approach for Aggressive Prostate Cancer during Active Surveillance

AS is commonly performed using a rigorous one-size-fits-all examination protocol to ensure safety. The monitoring protocol includes invasive procedures, such as prostate biopsies, imposing a burden on patients. The 5-year and 10-year continuation rates of AS are approximately 50% and 30%, respectively [15], implying that 50% of patients may undergo invasive examinations for up to five years, whereas 30% may endure them for a decade. Consequently, despite being a noninvasive treatment strategy, AS may subject some patients to active examinations, as it fails to consider the speed of individual disease progression. It enables a personalized risk-based follow-up that can identify aggressive PC upon enrolment for AS or during AS, minimizing unnecessary biopsies or treatments while promptly detecting progression. By utilizing personalized risk-based follow-ups, healthcare providers and patients can better understand the risks and make informed decisions regarding repeat biopsies [76]. This approach has been highlighted as a critical research priority at expert consensus conferences [77]. 

A review paper introduced risk calculators based on various clinical characteristics [76]. These calculators incorporate factors such as age, the PSA level, the PSA kinetics, the biopsy results, and the MRI findings, all of which show promising results [77,78,79,80,81,82]. Additionally, genetic testing may influence the risk classification and selection for AS. Integrating these genetic factors into individual genetic risk scores enables more accurate patient assessments [55,83]. The development of risk calculators based on clinical parameters, biomarkers, and genetic markers has the potential to enhance the diagnostic accuracy of aggressive PC during AS and reduce excessive testing.

However, there are doubts regarding the necessity of identifying aggressive PC in AS. AS primarily targets indolent PC cases. AS treatment involves deferring or avoiding definitive treatments with potential side effects. Thus, as long as the timing of curative interventions is not missed, despite the presence of aggressive PC, the goal of AS is achieved. Furthermore, patients who undergo an RP after AS for low-risk PC have favorable oncological outcomes, indicating that AS is sufficient for curative intent, even if performed before a definitive treatment [84,85]. Therefore, in recent years, as the indications for AS have expanded to intermediate-risk PC, the importance of identifying aggressive PC through a personalized risk-based follow-up is expected to increase.

## 6. Conclusions

AS is strongly recommended for low-risk PC, and is a promising approach for managing favorable intermediate-risk PC. It aims to minimize overtreatment and its associated adverse effects, while maintaining the health-related QOL. However, concerns remain regarding the identification of “aggressive PC” within the AS cohort, which refers to cancers that exhibit a higher potential for progression. The detection of aggressive PC during AS is crucial for timely interventions and appropriate follow-up strategies to improve the QOL of the patient.

To address these concerns, a personalized risk-based follow-up approach has been proposed to efficiently identify aggressive PC in patients undergoing AS. This approach involves the integration of clinical data, biomarkers, and genetic factors using risk calculators, allowing for a more accurate risk assessment and the early detection of disease progression. By implementing this risk-based follow-up system, unnecessary biopsies and treatments can be minimized and aggressive PC can be promptly addressed, leading to improved patient management and outcomes.

## Figures and Tables

**Table 1 cancers-15-04270-t001:** Recommendations for each guideline in active surveillance.

Guideline	Inclusion Criteria					Strength of Recommendation
	Clinical Stage	Gleason Score	Cancer Volume	PSA (ng/mL)	PSA Density (ng/mL/cm^3^)	
AUA [8]	≦T2	≦6	-	<20	-	strong
	≦T2	3 + 4	low % of pattern 4; <50% of total cores positive	<10	low	strong
EAU [9]	≦T2a	≦6	-	<10	-	strong
	≦T2a	3 + 4	<10% pattern 4; ≤3 cores positive; and ≤50% core involvement per core	<10	-	weak
NCCN [10]	≦T2a	≦6	-	<20	≦0.15	-
	≦T2a	3 + 4	low % of pattern 4; <50% of total cores positive	<10	low	-
JUA [17]	≦T2	≦6	≦2	<10	<0.2	grade B
	≦T2	3 + 4	≦2	<10	<0.2	weak

PSA, prostate-specific antigen; AUA, American Urological Association; EAU, European Association of Urology; NCCN, National Comprehensive Cancer Network; JUA, Japanese Urological Association.

**Table 2 cancers-15-04270-t002:** The follow-up protocols for various guidelines.

Guideline	Follow-Up Schedules			
	PSA	DRE	MRI	Repeat Biopsy
AUA [8]	not more frequently than every 6 months	every 1–2 years	increase in serial PSA, new DRE abnormalities, or clinical progression	increase in serial PSA, new DRE abnormalities, or clinical progression
EAU [9]	every 6 months	every 12 months	-	every 3 years for 10 years
NCCN [10]	every 6 months	every 12 months	every 12 months;	every year
JUA [17]	for first 2 years: every 3 months 3rd year onwards: every 6 months	every 6 months	additional MRI if not performed at diagnosis *, if PSA DT <10 years **, or prior to repeat biopsy **	at 1 year after diagnosis, every 3 years thereafter, and every 5 years after 10 years

PSA, prostate-specific antigen; DRE, digital rectal examination; MRI, magnetic resonance imaging; PSA-DT, PSA doubling time. * Perform targeted biopsy if the positive region is present to confirm that it meets eligibility criteria; ** perform targeted biopsy and systematic biopsy if the positive region is present.

## Data Availability

The data can be shared up on request.

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
