# Peer review of "Aggressive Prostate Cancer in Patients Treated with Active Surveillance"

_cancers, 2023, doi:10.3390/cancers15174270_

Round 1

Reviewer 1 Report

Congratulations to the authors on the present review. This is a relatively hot topic as the reach of AS keeps expanding and it is our reposnability to that safely and considering the risks and how to prevent them, to the best of our ability.

Author Response

Thank you for your comprehensive review and comments. We also feel that it is our responsibility to assess safety and consider and prevent risks.

Reviewer 2 Report

The manuscript is well written but some points should be improved

1) the role of transperineal prostate biopsy approach (Zattoni F, et al: Eur Urol Focus. 2023 Jul;9(4):621-628. doi: 10.1016/j.euf.2023.01.016. Epub 2023 Feb 4. PMID: 36746729) and number of needle cores (Pepe P, et al: Scand J Urol. 2017 Aug;51(4):260-263. doi: 10.1080/21681805.2017.1313310. Epub 2017 May 17. PMID: 28513296)  to reduce the risk of upgrading in men enrolled in AS should be added in the references

2) The role of genetic Markers to reduce the risk of upgrading should be improved in the text 

3) The Tables need to be simplified

4) The emergent role of PSMA PET/CT in the diagnosis of prostate cancer (Pepe P, et al Arch Ital Urol Androl. 2022 Sep 26;94(3):274-277. doi: 10.4081/aiua.2022.3.274. PMID: 36165469) and the possible role in the evaluation of men enrolled in active surveillance should (especially when mpMRI could not be performed) be added in the text

5) The role of PSAdensity to diagnose clinically significant PCa in men with negative mpMRI should be added in the text 

6) In the conclusions the sentence "AS has emerged as a promising approach for managing low-and favorable intermediate .....) should be rewritten because for all International prostate cancer guidelines AS is strongly recommend in low risk PCa

Author Response

I attached the file about the response to the reviewer.

Reviewer 3 Report

The paper is a narrative review about identification and management of aggressive prostate cancer during active surveillance. The topic is thoroughly examined. There are only two issues which need to be discussed.

1) MpMRI is yet mandatory before and during AS. The main role of MpMRI is not the reclassification of a tumour in regard to diffusion characteristic (at least at the moment being). Indeed, it may show an increased volume of the index lesion or may identify the onset of de novo lesion during AS. Both events may be linked to biological modification to the tumour leading to more aggressive PC on rebiopsy.

2) Anterior tumors are usually found by MpMRI and subsequent targeted biopsy. The biopsy may show a low volume, low grade tumor whereas, frequently, they are aggressive cancers.

Author Response

(The authors gave the same response as above.)

Reviewer 4 Report

This is a well written review manuscript on AS in prostate cancer. All aspects related to AS from the clinic to basic approach are reviewed, with special emphasis in the context of detecting PCA with aggressive fearures during AS. An in deep comparison of approached from different institutions an societies is provided. The reference list is updated and tables are illustrative.

Author Response

 We thank you for your time and effort invested in reviewing our manuscript, and greatly appreciate your comprehensive review and comments. The updated reference list and illustrative tables aim to enhance the manuscript's quality. Your comments are truly appreciated.

Round 2

Reviewer 2 Report

The manuscript has been improved